# Altered Gut Microbiota Patterns in Young Children with Recent Maltreatment Exposure

**DOI:** 10.3390/biom14101313

**Published:** 2024-10-16

**Authors:** Gergana Karaboycheva, Melanie L. Conrad, Peggy Dörr, Katja Dittrich, Elena Murray, Karolina Skonieczna-Żydecka, Mariusz Kaczmarczyk, Igor Łoniewski, Heiko Klawitter, Claudia Buss, Sonja Entringer, Elisabeth Binder, Sibylle M. Winter, Christine Heim

**Affiliations:** 1Institute of Medical Psychology, Charité–Universitätsmedizin Berlin, Corporate Member of Freie Universität Berlin and Humboldt-Universität zu Berlin, Berlin, Germany; gergana.karaboycheva@charite.de (G.K.); conradml@gmail.com (M.L.C.); heiko.klawitter@charite.de (H.K.); claudia.buss@charite.de (C.B.); sonja.entringer@charite.de (S.E.); 2Department of Child & Adolescent Psychiatry, Charité–Universitätsmedizin Berlin, Corporate Member of Freie Universität Berlin and Humboldt-Universität zu Berlin, Berlin, Germany; peggy.doerr@vivantes.com (P.D.); katja.dittrich@outlook.com (K.D.); elena.m.murray@gmail.com (E.M.); sibylle.winter@charite.de (S.M.W.); 3Department of Microbiology, Infectious Diseases and Immunology, Charité–Universitätsmedizin Berlin, Corporate Member of Freie Universität Berlin and Humboldt-Universität zu Berlin, Berlin, Germany; 4Department of Biochemical Research, Pomeranian Medical University, Szczecin, Poland; karolina.skonieczna@sanprobi.pl (K.S.-Ż.); mariusz.kaczmarczyk@sanprobi.pl (M.K.); igor.loniewski@pum.edu.pl (I.Ł.); 5Department of Pediatrics, University of California Irvine, Irvine, CA, USA; 6Max-Planck Institute of Psychiatry, Munich, Germany; binder@psych.mpg.de; 7Charité–Universitätsmedizin Berlin, Corporate Member of Freie Universität Berlin and Humboldt-Universität zu Berlin, NeuroCure Cluster of Excellence, Berlin, Germany

**Keywords:** childhood maltreatment, early life adversity, psychopathology, behavior, developmental programming, intestinal microbiota, gut–brain axis

## Abstract

Background: The brain and the intestinal microbiota are highly interconnected and especially vulnerable to disruptions in early life. Emerging evidence indicates that psychosocial adversity detrimentally impacts the intestinal microbiota, affecting both physical and mental health. This study aims to investigate the gut microbiome in young children in the immediate aftermath of maltreatment exposure. Methods: Maltreatment exposure was assessed in 88 children (ages 3–7) using the Maternal Interview for the Classification of Maltreatment [MICM]. Children were allocated to three groups according to the number of experienced maltreatment categories: no maltreatment, low maltreatment, and high maltreatment exposures. Stool samples were collected and analyzed by 16S rRNA sequencing. Results: Children subjected to high maltreatment exposure exhibited lower alpha diversity in comparison to those with both no and low maltreatment exposure (Simpson Index, Tukey post hoc, *p* = 0.059 and *p* = 0.007, respectively). No significant distinctions in beta diversity were identified. High maltreatment exposure was associated with the enrichment of several genera from the class Clostridia (*Clostridium, Intestinibacter, Howardella* and *Butyrivibrio*) and the depletion of the genus *Phocaeicola* (class Bacteriodia). Conclusions: Severe maltreatment exposure is associated with alterations in the gut microbiota of young children. Longitudinal trajectories of intestinal microbiota composition in the context of maltreatment may reveal important insights related to psychiatric and somatic health outcomes.

## 1. Introduction

Childhood maltreatment is a severe and common form of early life stress that is defined by exposure to physical violence, emotional or sexual abuse or any severe form of physical or emotional neglect [1,2,3]. Childhood maltreatment significantly increases the risk of stress-related disorders in adulthood, such as depression, anxiety, cardiovascular disease, metabolic disease and immune disorders [4,5,6,7,8]; however, the mechanisms for how early adversity developmentally programs mental health risk are still a matter of investigation. Essential pathways thought to play a role in adult disease risk with a history of childhood adversity include functional and structural changes in the limbic brain, dysregulation of the hypothalamus–pituitary–adrenal (HPA) stress axis and low-grade systemic inflammation [9,10,11,12,13].

We and others suggest that the microbiota–gut–brain (MGB) axis, understood as the bidirectional communication between the intestinal microbiota, the enteric immune and nervous systems and the brain, might represent an additional pathway mediating the impact of early adversity on disease risk [14,15,16,17]. The MGB axis is implicated in the acute stress response, anxiety, mood regulation and cognition due to close interactions between the intestinal microbiota and brain structures such as the HPA axis, the prefrontal cortex, the hippocampus, and the amygdala [18,19,20,21]. Several different pathways are thought to be involved in MGB signaling, including the vagus nerve, soluble immune mediators and metabolites produced by the intestinal flora [22,23,24]. Due to this connection, disruption of the gut microbiota due to severe stress exposure, especially during early life, may contribute to long-term dysfunction of the MBA axis [14,25].

Changes in the gut microbiota such as decreased bacterial diversity or alterations in the abundance and function of health-promoting or pathogenic bacteria (dysbiosis) have been associated with pathological phenotypes [26]. Both animal and human studies have shown that exposure to psychosocial stress is associated with a reduction in bacterial diversity [27,28,29,30], changes in the bacterial community profile [27,28,31,32,33,34], functional alterations in metabolite production [28,35,36] and increased inflammatory response [27,35,37]. Regarding preclinical research, fecal transplantation of gut bacteria from depressive patients into germ-free mice induced depression-like symptoms in the transplanted animals [38,39]. Additionally, germ-free models demonstrate that the absence of the gut microbiota significantly affected essential brain functions such as mood, social behavior, cognition and memory [40]. Of note, De Palma and colleagues showed that the presence of intestinal bacteria was critical for the manifestation of anxious behavior in animals exposed to the stress of maternal separation [41].

Childhood is a sensitive developmental period, in which the developing brain and gut microbiome are especially vulnerable to stress [25,42]. Early-life stress models in rodents demonstrated that psychosocial stress has implications for stress-related mood and behavior abnormalities [41,43,44]. Additional studies demonstrated that gut microbial dysbiosis accompanying early-life adversity, programmed HPA reactivity to early-life stressors [45]. Intriguingly, these effects were only reversible during early life and not in adulthood [45]. Finally, as part of the multi-hit or cumulative stress hypothesis, adverse health outcomes may be precipitated by a combination of several major negative events reaching allostatic overload, especially during developmental stages [46,47,48,49]. In accordance with this, early life is a critical window for early detection and intervention.

In children, examination of the intestinal microbiota in relation to experiences of childhood adversity revealed gut microbial dysbiosis in infants [50], toddlers [51], school children [52,53] and adolescents [54,55]. In these studies, however, the definition and assessment of adversity, as well as time since exposure, varied widely. Studies often relied on retrospective data or a broad range of adversity exposures, such as institutional care vs. being raised in a biological family [53,55], or a measure of negative events [51,54]. Only one study examined associations between more discrete predictors such as negative events, socioeconomic status, parent–child dysfunction and family turmoil [52]. Considering these points, more studies are needed to further our understanding of how early-life adversity affects the gut microbiome. The objective of this study is to assess how early-life maltreatment exposure influences the gut microbiota in 3- to 7-year-old children, and to test if cumulative differences in maltreatment differentially alter gut microbial composition. To our knowledge, this is the first study to examine the intestinal microbiota in young children with respect to a verified, in-depth assessment of maltreatment experiences.

## 2. Materials and Methods

### 2.1. Study Design

Study participants were enrolled in the Berlin Longitudinal Children Study (BerlinLCS, Berlin, Germany). Approval for the study was obtained from the ethics committee of Charité—Universitätsmedizin Berlin (Approval number EA2/161/13). Details of the recruitment strategy are available in the following publication [56].

A total of 89 children participated in the study as part of their last study visit within the BerlinLCS project, with 88 children included in the statistical analyses (one child was excluded from the no maltreatment group due to a traumatic experience during the course of the study). Participating children were aged between 3 and 7 years (Mean = 5.585, SD = 0.966). As part of the study, a full pediatric psychiatric anamnesis and a standard physical examination were conducted. None of the participating children suffered from any chronic disease. The last non-steroidal anti-inflammatory drug intake was at least a week prior to examination; the one child who used topical corticosteroids was included in the analysis. None of the participants used any other chronic medication.

### 2.2. Fecal Sample Collection

Fecal samples were collected either during the study visit or at home using a flushable toilet seat cover for stool collection and an OMNIgene.GUT home collection kit (DNA GENOTEK, Ottawa, ON, Canada). The DNA stabilization solution in this kit conserves bacterial DNA for up to 8 weeks at room temperature [57]. Stool samples collected at home were sent to the lab within a maximum of 2 weeks after collection and stored in the original collection tubes at −20°. The stool samples were processed and analyzed by the CEMET Center for Metagenomics (Tübingen, Germany). Stool DNA was extracted following the MetaHIT-protocol with a Stratec InviMag Stool DNA Kit (STRATEC Molecular GmbH, Birkenfeld, Germany) [58]. Amplicons from variable regions 3 and 4 of the 16s rRNA gene based on the primer 341F/785R were sequenced on a MiSeq Illumina sequencer [59]. The number of sequenced raw reads was above 50.000 in all samples. Quality trimming was performed by PRINSEQ 2 (version 0.20.4|0.13) [60], merging of short end reads by FLASH 3 (version 1.2.11) [61] and filtering of merged reads by PRINSEQ 2. The taxonomic assignment of bacterial DNA was conducted using the NCBI Bacterial 16S rRNA database, MALT 4 (version 0.6.3) [62] and the LCA algorithm [63].

### 2.3. Assessment of Maltreatment Exposure

Maltreatment exposure was assessed by trained interviewers using the Maternal Interview for the Classification of Maltreatment (MICM [64]), based on the Maltreatment Classification System (MCS [65]). The MCS classifies maltreatment into seven different categories (physical maltreatment, emotional maltreatment, physical neglect by insufficient care, physical neglect by insufficient supervision, sexual abuse, educational maltreatment, and moral maltreatment), and allows coding of the chronicity and severity of each exposure, for further details see [56]. Exposure to a larger number of different categories of maltreatment correlates with greater long-term negative psychological outcomes in adults [66,67] and appears to be highly associated with the severity of maltreatment exposure [68]. Thus, we chose as a predictor variable for our analyses cumulative maltreatment exposure measured by the number of maltreatment categories experienced over the course of life. Participants were allocated to a no-maltreatment-exposure group, a low-maltreatment-exposure group (1 to 2 maltreatment categories experienced) or a high-maltreatment-exposure group (3 or more maltreatment categories experienced). Exclusion criteria for the no-maltreatment-exposure group included any type of maltreatment (except for low-degree emotional maltreatment with a maximal severity score of 1 out of 5) as well as any other traumatic exposure or severe stressor.

### 2.4. Demographic Assessment

Information on child age, sex, ethnicity, socioeconomic status and antibiotic use was collected via a caregiver report. Socioeconomic status (SES) was assessed based on the Winkler and Stolzenberg Index [69,70,71]. This multi-dimensional index score, with a continuous variable range between 3 and 21 points, represents the sum of three metric components, i.e., education/occupational qualification, occupational status and household net income.

### 2.5. Assessment of Mental Health

Behavioral and emotional problems were rated using Child Behavior Checklist (CBCL) [72,73]. Depending on the child’s age, the 1.5–5-year or 6–18-year version of the questionnaire was used. We applied T-values based on United States norms for both versions as there are no published norm values for the German population for the 1.5–5-year version.

### 2.6. Statistical Analyses

All calculations were performed using R. Missing values in the SES (two cases) were substituted with the variable mean within the corresponding maltreatment exposure group. One missing value in the dummy variable “antibiotic use in the last 6 months” was treated by omission. Gut microbiome alpha diversity was calculated using the Shannon and Simpson indices [74] and computed using the Vegan package (version 2.6-8) [75]. To compare alpha diversity between groups, Analyses of Covariance (ANCOVA) were performed using the Car package [76]. Between-group differences were identified by Tukey post hoc comparisons. Two extreme outliers in the variable Simpson index (values below the threshold of 1st quartile+ 1.5 x interquartile range) were omitted from the ANCOVA analysis. Beta diversity was computed with the Bray–Curtis distance metric [77] using the Vegan package [75], and Principal Coordinate Analysis (PCoA) ordination was performed using the Phyloseq2 package (version 1.48.0) [78]. We compared beta diversity statistically among the three group means using Permutational Multivariate Analysis of Variance [79] with the Vegan package (“adonis2” function, 999 permutations, method “margin” for type III sum of squares). Differential abundance analyses were assessed using Negative Binomial Generalized Linear Models (NB-GLM) with *p*-value correction by the Benjamini–Hochberg Method using the Edge package [80]. Only common taxa present in more than 10% of the samples were included in the analyses of beta diversity and differential abundance. To explore the potential combined effects of maltreatment exposure and childhood mental health on the intestinal microbiota, we conducted interaction models for alpha and beta diversity using ANCOVA and PERMANOVA, respectively. We introduced the same covariates as in the original models described above. The interaction term included the variables maltreatment exposure and CBCL total score. Group comparisons regarding sample characteristics were performed using ANOVA and the Fisher exact test. The significance cut-off for adjusted *p* values was 0.05. In all analyses (including alpha diversity, beta diversity and differential abundance) age, sex, BMI, antibiotic intake in the previous 6 months and SES were included as covariates. In addition to differential abundance, relative abundance was calculated by dividing the read counts of each taxon by the total read count per sample. Heatmaps for each maltreatment group were generated to display the relative abundance of the most dominant taxa across samples.

## 3. Results

### 3.1. Sample Characteristics

Demographic information is presented in Table 1. In terms of age, sex or ethnicity, there were no significant differences between the no-exposure, low- or high-maltreatment-exposure groups. Children in the no-exposure group had significantly lower BMI than the low-exposure group, and children with both high and low exposure to maltreatment belonged to families with significantly lower socioeconomic status (SES) compared to children with no maltreatment exposure (*p* < 0.001).

### 3.2. Childhood Maltreatment Exposure Characteristics

The maltreatment exposure in our sample was characterized by an early onset within the first three years of life (mean 5.474, SD 9.714 months), high chronicity (mean 80.026% of the lifetime, SD 20.591%) and predominantly moderate severity (mean maximal severity 3.078, SD 0.941). Exposure to maltreatment was recent, as 83.78% of children experienced at least one category of maltreatment in the last 6 months before examination, with a mean 3.135, SD 6.754 months since the last exposure. The low- and high-maltreatment-exposure groups did not differ in terms of maximal experienced severity (most severe maltreatment event—lifetime) and recency (months since last exposure). However, the high-maltreatment-exposure group had a significantly higher cumulative severity (sum score of all experienced maltreatment events—lifetime, *p* = 0.001), and higher lifetime chronicity (total months of exposure/total months of life, *p* = 0.026), as shown in Table 2.

As to the different categories of maltreatment exposure, physical neglect (both in the form of insufficient care and insufficient supervision) as well as physical maltreatment were significantly more frequent among the children in the high-exposure group vs. the low-exposure group (see Table 2). One hundred percent of the children in both the low- and high-exposure groups had experienced emotional maltreatment. Sexual abuse was present in only three cases in the high-exposure group. Educational and moral/legal maltreatment were rare in our sample with only one case per category in the high maltreatment group. Within the no-exposure group, there was no-maltreatment-exposure except for one case of emotional maltreatment lifetime with low severity, in accordance with the inclusion criteria.

### 3.3. Gut Microbiota Analysis Reveals Reduced Alpha Diversity in Children with High-Maltreatment-Exposure Compared with Low-Maltreatment-Exposure Groups

To assess if maltreatment in early life affected the gut microbiota, we performed 16s rRNA sequencing on fecal samples from children in the BerlinLCS study. ANCOVA analysis of gut microbial alpha diversity showed a significant effect of maltreatment exposure (any) compared to the no-exposure group for the Simpson index (F (79,2) = 5.440, *p* = 0.006), as well as a trend for the Shannon index (F (80,2) = 2.974, *p* = 0.057), shown in Table 3a,b. Considering the Simpson index, Tukey post hoc analysis showed a significantly lower alpha diversity in the high-maltreatment-exposure group compared to the low-exposure group (t = −3.138, *p* = 0.007) and a trend compared to the no-exposure group (t = −2.322, *p* = 0.059), shown in Table 3c and Figure 1. There was no difference in alpha diversity between the low- and no-maltreatment-exposure groups. PERMANOVA analysis of gut microbial beta diversity with maltreatment exposure as the predictor, and age, BMI, sex, SES and antibiotic used as covariates revealed no significant differences between the maltreatment and no-exposure groups on a Bray–Curtis beta diversity distance metric (Table 4).

### 3.4. Maltreatment Exposure During Childhood Is Associated with Significant Alterations in Gut Microbial Composition

Next, we investigated the effect of maltreatment exposure on the gut microbial composition in children. Using an NB-GLM model, we demonstrate that low maltreatment exposure was associated with enrichment of the bacterial genera *Haemophilus*, *Eggerthella, Intestinibacter, Streptococcus* and *Sphingobium,* as well as depletion of the genus *Phocaeicola,* compared to the no-exposure group, as shown in Table 5, Figure 2A, relative abundance reported in Appendix A. Furthermore, a comparison of the high maltreatment with the no-exposure group revealed the enrichment of *Howardella, Veillonella, Intestinibacter, Schaalia*, *Butyrivibrio* and *Clostridium*, as well as depletion of *Agathobaculum and Phocaeicola*, as shown in Table 6 and Figure 2B, relative abundance shown in Appendix A. Similarities between the low- and high-maltreatment groups include enrichment of *Intestinibacter*, as well as depletion of *Phocaeicola.*

### 3.5. Childhood Emotional and Behavioral Problems Show No Significant Effects on Intestinal Alpha and Beta Diversity

Finally, we examined the potential combined effects of maltreatment exposure and childhood emotional and behavioral problems on intestinal alpha and beta diversity. The ANCOVA and PERMANOVA interaction models for alpha and beta diversity respectively found no significant effects of CBCL total score or interaction with maltreatment exposure. The result tables are provided in Appendix A.

## 4. Discussion

Our study revealed that the gut microbiome of children with high maltreatment exposure showed significantly lower alpha diversity compared to children from the low-maltreatment-exposure group and a trend compared to children from the no-exposure group. The relationship between maltreatment exposure and alpha diversity did not vary based on the number of mental health symptoms in the children. A diverse gut microbiome (i.e., high alpha diversity) is important for the proper development of the immune system and, subsequently, the gut–brain axis. Lower alpha diversity has been linked to increased inflammation, which can negatively impact brain development and mental health outcomes. Additionally, maltreatment exposure was associated with enrichment of multiple genera from the class Clostridia including *Intestinibacter, Howardella, Clostridium* and *Butyrivibrio* as compared to the no-exposure group. *Intestinibacter* was enriched in both the low- and high-maltreatment groups, whereas *Phocaeicola* was depleted. The observed changes to *Clostridia* are notable as this could be a potential attempt at protecting the gut–brain axis through the production of anti-inflammatory metabolites. These data demonstrate that exposure to maltreatment in early childhood is associated with decreased alpha diversity and changes to gut microbial composition, with marked enrichment in bacterial clades from the class Clostridia.

Psychosocial stress has been consistently associated with lower alpha diversity in human cohorts, and our findings from the BerlinLCS cohort agree with the literature. Callaghan et al. observed lower richness in the gut microbiota of children aged 5 to 11 who were exposed to institutional care [53], and postnatal adversity in 2-year-old children was associated with lower phylogenetic diversity [51]. Additionally, 8–16-year-old adolescents who experienced a high number of negative psychosocial events also showed decreased alpha diversity compared to a control group [54]. In addition to the young age of the children, the stratification into low- and high-maltreatment-exposure groups is a novel aspect of our study. Indeed, our observation of decreased diversity only in the high maltreatment group suggests that cumulative or multi-hit exposure in terms of increasing allostatic load may play a role in shaping intestinal alpha diversity. Evidence suggests that exposure to a larger number of different types of maltreatment (multitype maltreatment) is associated with higher maltreatment severity and greater long-term negative psychological outcomes in adults [66,67]. Accordingly, in our study, children with higher cumulative maltreatment exposure had experienced a higher cumulative severity of maltreatment, had more chronic exposure, and showed more pronounced microbiota alterations than children with low maltreatment exposure.

Considering gut bacterial composition in our sample, the high-maltreatment-exposure group exhibited enrichment in the genus *Clostridium* compared to the no-maltreatment group. In animal models, an increased abundance of the genus *Clostridium* has been repeatedly reported in different psychological stress paradigms [27,81,82,83], as well as in offspring that experienced maternal separation in early life [41,84]. In humans, there is a paucity of studies examining the microbiome in young children after exposure to early life adversity, and a comparison of the existing literature shows conflicting results. For instance, prenatal adversity exposure was associated with significantly increased intestinal *Clostridium* in 2-year-old children [51]; however, a study of 8–16-year-old children, demonstrated that high stress was associated with decreased abundance of the phylum Firmicutes [54]. This highlights the critical need for longitudinal studies assessing the impact of maltreatment in early life on the microbiome, specifically focusing on this relationship closer to the time point of exposure.

A further aspect to consider is that specific subtypes of maltreatment might contribute to microbiota alterations in unique ways beyond the effect of increasing allostatic load in cumulative exposure. For example, the potential impact of diet and hygiene in maltreated children, particularly in cases involving physical neglect, or inflammatory processes in case of physical injury. Further exploration of these factors could shed light on additional dimensions of the complex interplay between environment, physiological stress responses, and gut microbial composition in the context of childhood maltreatment. Finally, protective mechanisms such as an increase in SCFA production that maintain gut homeostasis during the experience of stress might be at play parallel to dysbiotic trends and phenomena. In line with the hypothesis of allostatic load, dysbiosis and health disorders might manifest when cumulative negative hits outweigh the protective mechanisms of the system.

The current study is subject to several limitations. Firstly, the assessment lacks information on the children´s eating habits. Cumulative exposure to adverse childhood experiences as well as low SES are predictors for obesity and poorer diet [85,86,87,88,89]. For instance, populations with low SES, including children and adolescents, are more prone to a diet rich in refined grains, sugar and added fats, and low in whole vegetables and fiber compared to populations with high SES [90,91,92]. Intake of whole fruits, vegetables and fiber is associated with higher bacterial richness and lowered ratio of abundance of Firmicutes to Bacteroidetes (F:B), as well as with a lower abundance of potential pathogens [93]. While consistent results on the effects of sweeteners on gut microbiota are missing [94], data from animal studies have suggested that emulsifiers, which are often found in processed foods, can lead to an increase in the abundance of the clades *Veionella* and *Clostridium* [95]. Despite controlling our results for BMI and SES, the observed differences in intestinal microbiota patterns between children with high, low and no maltreatment exposure might still be influenced by diet and should be interpreted with caution. Further, some environmental factors, which are also known to additionally shape the gut microbiota, such as physical activity, sleep, pets and home hygiene, could not be assessed in our cohort.

Another limitation consists of the cross-sectional study design, which does not reflect the dynamic fluctuations in time, which are characteristic of the intestinal microbiota composition. The results from this study, however, provide important indications for the success of longitudinal studies. Finally, due to the relatively small sample size, it was not possible to examine the effects of specific maltreatment categories on the microbiota composition, and thus to differentiate the effects of multi-hit exposure vs. the influence of a specific type of maltreatment exposure.

This study shows that cumulative maltreatment exposure is associated with a dysbiotic intestinal microbiota profile in children. Since dysregulation of the microbiome–gut–brain axis can contribute to both somatic and psychiatric pathology during the life course, this remains an important area of study. Future studies should aim to create longitudinal observation frameworks to examine the mechanisms of how lifestyle and stress exposure affect intestinal microbiota and developmental trajectories along the gut–brain axis in childhood. Such studies could advance the development of novel early interventions targeting the gut microbiota in early life.

## Figures and Tables

**Figure 1 biomolecules-14-01313-f001:**
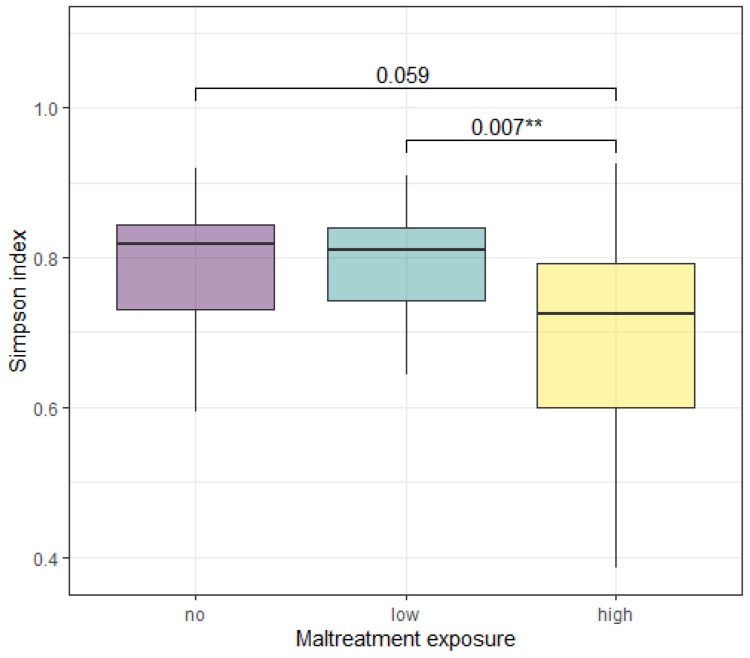
Alpha diversity in the no-, low- and high-maltreatment-exposure groups, according to the Simpson Index. The high-maltreatment-exposure group shows a significantly lower alpha diversity compared to the no- and low-maltreatment-exposure groups. ANCOVA with post hoc Tukey analysis. Significance is represented by ** *p* < 0.01.

**Figure 2 biomolecules-14-01313-f002:**
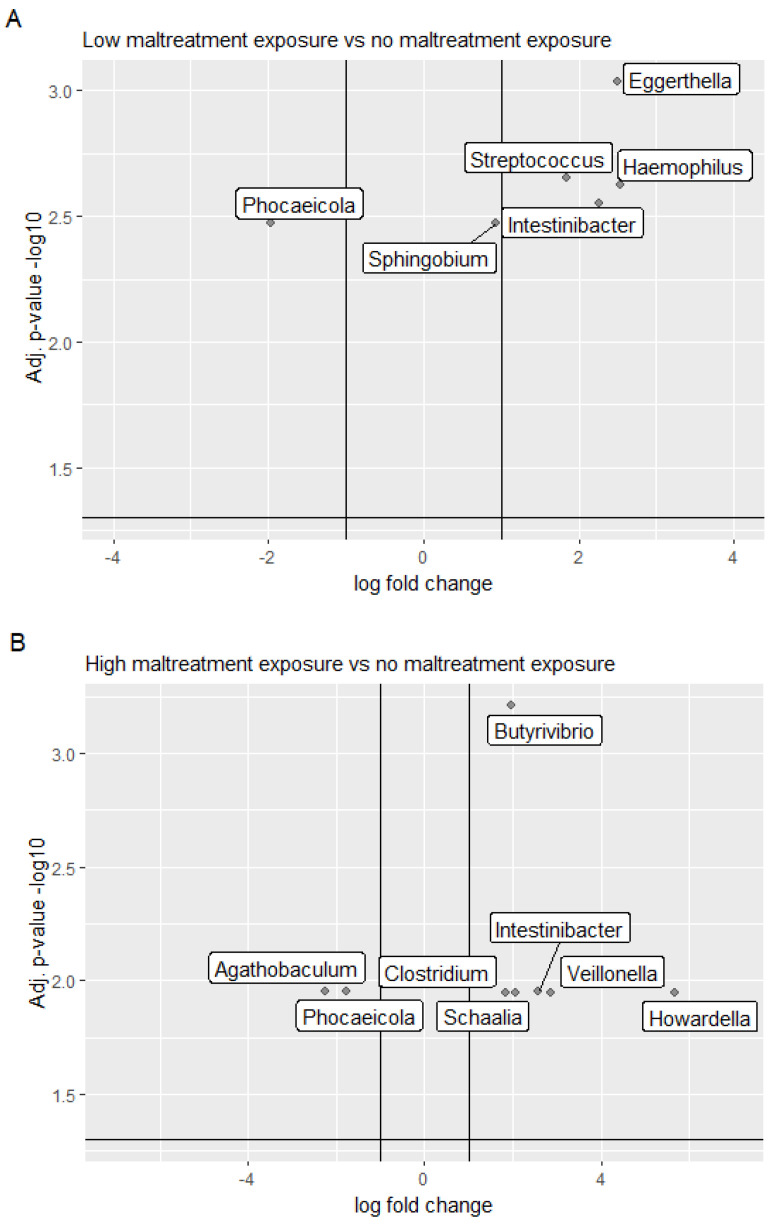
Volcano plots representing bacterial differential abundance between (**A**) the low- and the no-maltreatment-exposure group, and (**B**) the high and the no-maltreatment-exposure group. The horizontal line marks the FDR adjusted *p* = 0.05 cut off. The vertical lines mark a log fold change of 1 and −1, respectively. Positive log fold change values indicate taxa that are relatively more abundant in the multitype maltreatment group while negative log fold change values taxa, which are relatively depleted. We have labeled only genera with significant differential abundance.

**Table 1 biomolecules-14-01313-t001:** Subject characteristics (no maltreatment exposure, low and high maltreatment exposure). ANOVA with Tukey post hoc or Fischer Exact Test. Exp. = exposure, n.s. = not significant.

	0. No Exp. *n* = 50 Mean (SD) or *n* (%)	1. Low Exp. *n* = 21 Mean (SD) or *n* (%)	2. High Exp. *n* = 17 Mean (SD) or *n* (%)	*p* Value
**Age (Years)**	5.440 (0.970)	5.736 (0.909)	5.825 (1.002)	n.s.
**Sex (Male)**	24 (48%)	12 (57%)	8 (47%)	n.s.
**Ethnicity** **(Caucasian)**	48 (96%)	18 (86%)	15 (88%)	n.s.
**BMI**	14.928 (1.061)	15.895 (1.512)	15.347 (1.127)	**1 vs. 0: 0.007** 2 vs. 0: n.s. 2 vs. 1: n.s.
**SES**	17.100 (3.500)	9.050 (5.145)	10.250 (4.450)	**1 vs. 0: <0.001** **2 vs. 0: <0.001** 2 vs. 1: n.s.

**Table 2 biomolecules-14-01313-t002:** Comparison of characteristics and categories between the low- and high-maltreatment-exposure groups. Student’s t test and Fischer Exact Test. n.s = not significant, NA = Not applicable.

Maltreatment Characteristics	Low-Exposure Mean (SD)	High-Exposure Mean (SD)	*p* Value
Age at first onset (Months)	8.429 (11.552)	1.824 (5.077)	**0.026**
Maximal severity (Lifetime)	3.000 (1.000)	3.176 (0.760)	n.s.
Cumulative severity (Lifetime)	9.524 (5.046)	16.118 (6.133)	**0.001**
Chronicity (Total months of exposure/total age in months)	0.720 (0.223)	0.898 (0.883)	**0.027**
Recency (Months since last exposure)	4.000 (6.743)	2.118 (6.827)	n.s.
**Maltreatment Categories**	**Low-Exposure** **Frequency (%)**	**High-Exposure** **Frequency (%)**	***p* Value**
Neglect insufficient supervision	3 (14.29)	10 (58.82)	**0.006**
Neglect insufficient care	2 (9.52)	15 (88.24)	**<0.001**
Physical maltreatment	7 (33.33)	12 (70.59)	**0.049**
Sexual abuse	0	3 (17.65)	NA
Emotional maltreatment	21 (100)	17 (100)	n.s.
Educational maltreatment	0	1 (5.88)	NA
Moral/legal maltreatment	0	1 (5.88)	NA

**Table 3 biomolecules-14-01313-t003:** Comparison of gut microbial alpha diversity with maltreatment exposure as a predictor and age, BMI, SES, sex and antibiotic use as covariates: (a) Shannon index. (b) Simpson index. (c) Simpson index post hoc analysis. ANCOVA and Tukey post hoc analysis. Significance is represented by ** *p* < 0.01.

(a) **Shannon index (ANCOVA)**	**F (df)**	***p* value**
Age	0.069 (79.1)	0.793
BMI	0.502 (79.1)	0.481
Sex	3.033 (79.1)	0.085
SES	0.410 (79.1)	0.524
Antibiotic use	0.514 (79.1)	0.476
Maltreatment exposure	2.974 (79.2)	0.057
(b) **Simpson index (ANCOVA)**	**F (df)**	***p* value**
Age	0.589 (79.1)	0.445
BMI	1.326 (79.1)	0.253
Sex	1.453 (79.1)	0.232
SES	0.505 (79.1)	0.479
Antibiotic use	0.101 (79.1)	0.752
Maltreatment exposure	5.440 (79.2)	**0.006 ****
(c) **Simpson index (Tukey post hoc)**	**t value**	***p* value**
Low exposure vs. no exposure	0.688	0.771
High exposure vs. no exposure	−2.322	0.059
High exposure vs. low exposure	−3.138	**0.007 ****

**Table 4 biomolecules-14-01313-t004:** Comparison of the gut microbiota revealed no differences in beta diversity between the no-exposure, low- or high-maltreatment-exposure groups. PERMANOVA with maltreatment exposure as a predictor, including age, BMI, sex, SES and antibiotic use as covariates. Significance is represented by * *p* < 0.05.

Gut Microbial Beta Diversity	F (df)	Adj. *p* Value
Age	2.533 (79.1)	0.029 *
BMI	1.906 (79.1)	0.094
Sex	1.694 (79.1)	0.157
SES	1.499 (79.1)	0.173
Antibiotic use	2.152 (79.1)	0.060
Maltreatment exposure	1.446 (79.2)	0.171

**Table 5 biomolecules-14-01313-t005:** Differentially abundant bacterial taxa in the low-maltreatment-exposure group relative to the no-maltreatment-exposure group (NB-GLM). Log Fold Change expresses the log-transformed ratio between counts of differentially abundant bacteria. g = genus, c = Class, LogFC = Log Fold Change, FDR = *p* value adjusted by the False Discovery Rate (FDR) method. Only significant taxa are displayed.

Low Maltreatment vs. No Maltreatment	
**Enriched Bacterial Clades**	**LogFC (FDR)**
*g_Haemophilus* (c_Gammaproteobacteria)	2.535 (0.002)
*g_Eggerthella* (c_Coriobacteriia)	2.497 (0.001)
*g_Intestinibacter* (c_Clostridia)	2.270 (0.003)
*g_Streptococcus* (c_Bacilli)	1.844 (0.002)
*g_Sphingobium* (c_Alphaproteobacteria)	0.936 (0.003)
**Depleted bacterial clades**	**LogFC (FDR)**
g_*Phocaeicola* (c_Bacteroidia)	−1.971 (0.003)

**Table 6 biomolecules-14-01313-t006:** Differentially abundant bacterial taxa in the high-maltreatment-exposure group compared to the no-maltreatment-exposure group (NB-GLM). Log Fold Change expresses the log-transformed ratio between counts of differentially abundant bacteria. g = genus, c = Class, LogFC = Log Fold Change, FDR = *p* value adjusted by the False Discovery Rate (FDR) method. Only significant taxa are displayed.

High Maltreatment vs. No Maltreatment	
**Enriched Bacterial Clades**	**LogFC (FDR)**
*g_Howardella* (c_ Clostridia)	5.671 (0.011)
*g_Veillonella* (c_Negativicutes)	2.870 (0.011)
*g_Intestinibacter* (c_Clostridia)	2.584 (0.011)
*g_Schaalia* (c_Actinomycetia)	2.079 (0.011)
*g_Butyrivibrio* (c_Clostridia)	1.955 (0.001)
*g_Clostridium* (c_Clostridia)	1.838 (0.011)
**Depleted bacterial clades**	**LogFC (FDR)**
*g_Agathobaculum* (c_Clostridia)	−2.241 (0.011)
*g_Phocaeicola* (c_Bacteriodia)	−1.762 (0.011)

## Data Availability

The data that support the findings of this study are available from the corresponding author upon reasonable request.

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
