# Peer review of "Altered Gut Microbiota Patterns in Young Children with Recent Maltreatment Exposure"

_biomolecules, 2024, doi:10.3390/biom14101313_

Round 1

Reviewer 1 Report

Comments and Suggestions for Authors

 The study by Karaboychev et al. "Altered gut microbiota patterns in young children with recent maltreatment exposure" investigates the impact of childhood adversity on gut microbiota composition in children. This is a very interesting contribution to the research on the link between early life adversity and gut microbiota. The experiment is well designed and clearly presented. I have only a minor concern that should be revised before publication and a few suggestions for improving the manuscript.

INTRODUCTION

Both animal and human studies have  shown that exposure to psychosocial stress is associated with reduction of bacterial diversity [27–29], changes in the bacterial community profile [27,28,30–33], functional alterations in metabolite production [28,32] and increased inflammatory response [27,32]”

For preclinical literature, more articles on the effects of early-life stress (e.g., the effects of maternal separation) on gut microbiota and gut barrier integrity should be cited rather than articles on adult stress and microbiota.

 ·       Salberg S, Macowan M, Yamakawa GR, Beveridge JK, Noel M, Marsland BJ, Mychasiuk R. Gut instinct: Sex differences in the gut microbiome are associated with changes in adolescent nociception following maternal separation in rats. Dev Neurobiol. 2023 Jul-Sep;83(5-6):219-233. doi: 10.1002/dneu.22925. Epub 2023 Jul 24. PMID: 37488954.

 ·       Collins JM, Caputi V, Manurung S, Gross G, Fitzgerald P, Golubeva AV, Popov J, Deady C, Dinan TG, Cryan JF, O'Mahony SM. Supplementation with milk fat globule membrane from early life reduces maternal separation-induced visceral pain independent of enteric nervous system or intestinal permeability changes in the rat. Neuropharmacology. 2022 Jun 1;210:109026. doi: 10.1016/j.neuropharm.2022.109026. Epub 2022 Mar 10. PMID: 35283136.

 ·       Rincel M, Olier M, Minni A, Monchaux de Oliveira C, Matime Y, Gaultier E, Grit I, Helbling JC, Costa AM, Lépinay A, Moisan MP, Layé S, Ferrier L, Parnet P, Theodorou V, Darnaudéry M. Pharmacological restoration of gut barrier function in stressed neonates partially reverses long-term alterations associated with maternal separation. Psychopharmacology (Berl). 2019 May;236(5):1583-1596. doi: 10.1007/s00213-019-05252-w. Epub 2019 May 23. PMID: 31147734.

 ·       Riba A, Olier M, Lacroix-Lamandé S, Lencina C, Bacquié V, Harkat C, Gillet M, Baron M, Sommer C, Mallet V, Salvador-Cartier C, Laurent F, Théodorou V, Ménard S. Paneth Cell Defects Induce Microbiota Dysbiosis in Mice and Promote Visceral Hypersensitivity. Gastroenterology. 2017 Dec;153(6):1594-1606.e2. doi: 10.1053/j.gastro.2017.08.044. Epub 2017 Sep 1. PMID: 28865734.

 “Of note, De Palma and colleagues showed that the presence of intestinal  bacteria was critical for the manifestation of anxious behavior in stressed animals [37].

It is important to specify that De Palma's work was carried out on animals exposed to the stress of maternal separation.

Finally, as part of the multi-hit or cumulative stress hypothesis, adverse  health outcomes may be precipitated by a combination of several major negative events reaching allostatic overload, especially during developmental stages [42,43].

Preclinical studies on the gut-brain axis using neonatal models with multiple hits a should be cited.

 Rincel M, Aubert P, Chevalier J, Grohard PA, Basso L, Monchaux de Oliveira C, Helbling JC, Lévy É, Chevalier G, Leboyer M, Eberl G, Layé S, Capuron L, Vergnolle N, Neunlist M, Boudin H, Lepage P, Darnaudéry M. Multi-hit early life adversity affects gut microbiota, brain and behavior in a sex-dependent manner. Brain Behav Immun. 2019 Aug;80:179-192. doi: 10.1016/j.bbi.2019.03.006. Epub 2019 Mar 11. PMID: 30872090.

 Katz-Barber MW, Hollins SL, Cuskelly A, Leong AJW, Dunn A, Harms L, Hodgson DM. Investigating the gut-brain axis in a neurodevelopmental rodent model of schizophrenia. Brain Behav Immun Health. 2020 Feb 13;3:100048. doi: 10.1016/j.bbih.2020.100048. PMID: 34589838; PMCID: PMC8474551.

  METHODS:

The method of assignment to the "low maltreatment" or "high assessment" group should be described more precisely. It is not clear that the number of categories of maltreatment makes it possible to distinguish between low and high levels of maltreatment, especially if we consider that the different categories may have different severities.

 RESULTS:

The percentage of subjects in each category (physical maltreatment, emotional maltreatment, physical neglect by insufficient care, physical neglect by insufficient supervision, sexual abuse, educational maltreatment, and moral maltreatment) should be provide in the subjects characteristics table.

 In Table 3, the authors should add age as a possible contributing factor. This is a key factor contributing to microbiota composition, particularly in young children.

 DISCUSSION:

An important limitation of the study is the lack of nutritional information. Given that there is a significant difference between the groups (no adversity vs. high adversity) for SES, we cannot exclude that nutrition also differs between these groups. This aspect should be discussed in more detail.

Author Response

Reviewer 1

We thank Reviewer 1 for their thoughtful comments, which have improved the manuscript in its revised form.

The study by Karaboychev et al. "Altered gut microbiota patterns in young children with recent maltreatment exposure" investigates the impact of childhood adversity on gut microbiota composition in children. This is a very interesting contribution to the research on the link between early life adversity and gut microbiota. The experiment is well designed and clearly presented. I have only a minor concern that should be revised before publication and a few suggestions for improving the manuscript.

INTRODUCTION

Question 1. “Both animal and human studies have shown that exposure to psychosocial stress is associated with reduction of bacterial diversity [27–29], changes in the bacterial community profile [27,28,30–33], functional alterations in metabolite production [28,32] and increased inflammatory response [27,32]”

For preclinical literature, more articles on the effects of early-life stress (e.g., the effects of maternal separation) on gut microbiota and gut barrier integrity should be cited rather than articles on adult stress and microbiota.

  • Salberg S, Macowan M, Yamakawa GR, Beveridge JK, Noel M, Marsland BJ, Mychasiuk R. Gut instinct: Sex differences in the gut microbiome are associated with changes in adolescent nociception following maternal separation in rats. Dev Neurobiol. 2023 Jul-Sep;83(5-6):219-233. doi: 10.1002/dneu.22925. Epub 2023 Jul 24. PMID: 37488954.
  • Collins JM, Caputi V, Manurung S, Gross G, Fitzgerald P, Golubeva AV, Popov J, Deady C, Dinan TG, Cryan JF, O'Mahony SM. Supplementation with milk fat globule membrane from early life reduces maternal separation-induced visceral pain independent of enteric nervous system or intestinal permeability changes in the rat. Neuropharmacology. 2022 Jun 1;210:109026. doi: 10.1016/j.neuropharm.2022.109026. Epub 2022 Mar 10. PMID: 35283136.
  • Rincel M, Olier M, Minni A, Monchaux de Oliveira C, Matime Y, Gaultier E, Grit I, Helbling JC, Costa AM, Lépinay A, Moisan MP, Layé S, Ferrier L, Parnet P, Theodorou V, Darnaudéry M. Pharmacological restoration of gut barrier function in stressed neonates partially reverses long-term alterations associated with maternal separation. Psychopharmacology (Berl). 2019 May;236(5):1583-1596. doi: 10.1007/s00213-019-05252-w. Epub 2019 May 23. PMID: 31147734.
  • Riba A, Olier M, Lacroix-Lamandé S, Lencina C, Bacquié V, Harkat C, Gillet M, Baron M, Sommer C, Mallet V, Salvador-Cartier C, Laurent F, Théodorou V, Ménard S. Paneth Cell Defects Induce Microbiota Dysbiosis in Mice and Promote Visceral Hypersensitivity. Gastroenterology. 2017 Dec;153(6):1594-1606.e2. doi: 10.1053/j.gastro.2017.08.044. Epub 2017 Sep 1. PMID: 28865734.

Answer 1: Thank you for this comment that has improved the introduction. We have added the references on Page 2, Lines 72-74.

Question 2. Of note, De Palma and colleagues showed that the presence of intestinal bacteria was critical for the manifestation of anxious behavior in stressed animals [37]. “

It is important to specify that De Palma's work was carried out on animals exposed to the stress of maternal separation.

Answer 2: Thank you for catching this. We have altered the text as follows on Page 2, Lines 79-81.

“Of note, De Palma and colleagues showed that the presence of intestinal bacteria was critical for the manifestation of anxious behavior in animals exposed to the stress of maternal separation [37].”

Question 3. Finally, as part of the multi-hit or cumulative stress hypothesis, adverse health outcomes may be precipitated by a combination of several major negative events reaching allostatic overload, especially during developmental stages [42,43].

Preclinical studies on the gut-brain axis using neonatal models with multiple hits a should be cited.

 Rincel M, Aubert P, Chevalier J, Grohard PA, Basso L, Monchaux de Oliveira C, Helbling JC, Lévy É, Chevalier G, Leboyer M, Eberl G, Layé S, Capuron L, Vergnolle N, Neunlist M, Boudin H, Lepage P, Darnaudéry M. Multi-hit early life adversity affects gut microbiota, brain and behavior in a sex-dependent manner. Brain Behav Immun. 2019 Aug;80:179-192. doi: 10.1016/j.bbi.2019.03.006. Epub 2019 Mar 11. PMID: 30872090.

 Katz-Barber MW, Hollins SL, Cuskelly A, Leong AJW, Dunn A, Harms L, Hodgson DM. Investigating the gut-brain axis in a neurodevelopmental rodent model of schizophrenia. Brain Behav Immun Health. 2020 Feb 13;3:100048. doi: 10.1016/j.bbih.2020.100048. PMID: 34589838; PMCID: PMC8474551.

Answer 3: We have added these references on Page 2, Line 90.

METHODS:

Question 4. The method of assignment to the "low maltreatment" or "high assessment" group should be described more precisely. It is not clear that the number of categories of maltreatment makes it possible to distinguish between low and high levels of maltreatment, especially if we consider that the different categories may have different severities.

Answer 4: We have further described the reasoning behind choosing the predictor variable and extended the description of the low and high maltreatment groups with the following text, found on Page 3-4, Lines 146-156.

“Exposure to a larger number of different categories of maltreatment correlates with greater long-term negative psychological outcomes in adults [60,61] and appears to be highly associated with the severity of maltreatment exposure [62]. Thus, we chose as predictor variable for our analyses cumulative maltreatment exposure measured by the number of experienced of maltreatment categories over the course of life. Participants were allocated to a no maltreatment exposure group, a low maltreatment exposure group (1 to 2 maltreatment categories experienced) or a high maltreatment exposure group (3 or more maltreatment categories experienced). Exclusion criteria for the no maltreatment exposure group included any type of maltreatment (except for low degree emotional maltreatment with a maximal severity score of 1 out of 5) as well as any other traumatic exposure or severe stressor. As shown in Table 2, in our sample too the exposure to higher number of maltreatment categories was associated with higher cumulative severity, higher chronicity and earlier onset by similar recency of exposure.”

 RESULTS:

Question 5: The percentage of subjects in each category (physical maltreatment, emotional maltreatment, physical neglect by insufficient care, physical neglect by insufficient supervision, sexual abuse, educational maltreatment, and moral maltreatment) should be provide in the subjects characteristics table.

Answer 5: We have added this information to Table 2 on Page 5 as requested, and also added the following information to the results section on Page 5, Lines 223-231.

“As to the different categories of maltreatment exposure, physical neglect (both in form of insufficient care and insufficient supervision) as well as physical maltreatment were significantly more frequent among the children in the high exposure group vs the low exposure group (see Table 2). 100% of the children in both the low and high exposure groups had experienced emotional maltreatment. Sexual abuse was present in only 3 cases in the high exposure group. Educational and moral- legal maltreatment resulted to be rare in our sample with only one case per category in the high maltreatment group. Within the no exposure group there was no maltreatment expo-sure except for one case of emotional maltreatment lifetime with low severity, in accordance with the inclusion criteria.”

Question 6. In Table 3, the authors should add age as a possible contributing factor. This is a key factor contributing to microbiota composition, particularly in young children.

Answer 6: We have added age as a covariate for alpha and beta diversity analyses as well as differential abundance. We have revised the manuscript as follows: Methods: Page 4 Lines 197-199. Results:  Table 3 (Page 6), Table 4 (Page 7), Table 5 (Page 8), Table 6 (Page 8).

 DISCUSSION:

Question 7. An important limitation of the study is the lack of nutritional information. Given that there is a significant difference between the groups (no adversity vs. high adversity) for SES, we cannot exclude that nutrition also differs between these groups. This aspect should be discussed in more detail.

Answer 7: We agree that nutrition contributes to the composition of the gut microbiome. As we unfortunately do not have nutritional information for this study we have included this information in the limitations, found in the Discussion on Page 11, Lines 364-366.

“Further, some environmental factors, which are known to additionally shape the gut microbiota, such as diet, physical activity, sleep, pets and home hygiene, could not be assessed in our cohort.”

Reviewer 2 Report

Comments and Suggestions for Authors

Karaboycheva and colleagues report that children ages 3-7 with recent maltreatment exposure demonstrate an associated alteration to their gut microbiota. This is a novel and interesting study, given the strong connections between early life development on long-term outcomes and implications of the microbiota-gut-brain axis. I have major concerns about the reporting and interpretation of data, especially the gut microbiota data for the authors to consider to improve this manuscript.

-    The authors include antibiotic intake from the past 6 months in their analyses, as antibiotics are known to impact microbiome composition. The analysis would be greatly strengthened by including information about diet, as diet overwhelmingly affects gut microbiota composition. This could be via food records or estimated by proxy using one of the maltreatment criteria.

-            Differentially abundant taxa contain some unusual species. For example, Caloramator appears to be an environmental bacterium that is not well-characterized, as is Howardella being isolated from sheep rumen. In Figure 2A there are more unusual species, like Vallitalea which has been isolated from marine sediment. This raises concerns over the microbiome analysis, whether this is from their bioinformatics pipeline or contamination of samples. It is also highly concerning and unusual that relative abundance data is not shown. Running differential abundance analysis on bacterial counts is not informative, especially when differences in Simpson index diversity were presently reported (which takes relative abundance into account)

-            Lines 327-329: the authors make connections between Clostridia and HPA activities based on other studies. Reference 74 (Zaytsoff et al) states that corticosterone is mediated specifically by Clostridium perfringens, which is from the Clostridia class. However authors the authors erroneously connect this to their observed increase in bacterial taxa within the Clostridia class, but they do not report increases in Clostridium perfringens in the current study. Species within the same family, order, phylum etc. may have vastly different functions and I advise the authors to reassess their interpretation of microbiome data.

Author Response

Reviewer 2

We thank reviewer 2 for their time in reviewing and commenting on our manuscript.

Karaboycheva and colleagues report that children ages 3-7 with recent maltreatment exposure demonstrate an associated alteration to their gut microbiota. This is a novel and interesting study, given the strong connections between early life development on long-term outcomes and implications of the microbiota-gut-brain axis. I have major concerns about the reporting and interpretation of data, especially the gut microbiota data for the authors to consider to improve this manuscript.

Question 1. The authors include antibiotic intake from the past 6 months in their analyses, as antibiotics are known to impact microbiome composition. The analysis would be greatly strengthened by including information about diet, as diet overwhelmingly affects gut microbiota composition. This could be via food records or estimated by proxy using one of the maltreatment criteria.

Answer 1: We completely agree with this assessment, however, unfortunately do not have nutritional information for this study. As stated in our answer to Reviewer 1, Comment 7, since we unfortunately do not have nutritional information for this study we have included this information in the limitations, found in the Discussion on Page 11, Lines 345-366.

“Further, some environmental factors, which are known to additionally shape the gut microbiota, such as diet, physical activity, sleep, pets and home hygiene, could not be assessed in our cohort.”

Question 2: Differentially abundant taxa contain some unusual species. For example, Caloramator appears to be an environmental bacterium that is not well-characterized, as is Howardella being isolated from sheep rumen. In Figure 2A there are more unusual species, like Vallitalea which has been isolated from marine sediment. This raises concerns over the microbiome analysis, whether this is from their bioinformatics pipeline or contamination of samples. It is also highly concerning and unusual that relative abundance data is not shown. Running differential abundance analysis on bacterial counts is not informative, especially when differences in Simpson index diversity were presently reported (which takes relative abundance into account)

Answer 2: Thank you for catching this error regarding the unusual species. We have since reanalyzed the data and the revised information can be found in Table 5 (Page 8) Table 6 (Page 8) and Figure 2 (Page 9). Additionally we have revised the Abstract on Page  1, Lines 36-41, Results on Page 8, Lines 264-272, Discussion on Page 10, Lines 308-312. Regarding the genus Howardella, which we found increased differential abundance in the High Maltreatment versus No Maltreatment group, this bacteria is indeed found in the human intestine 1-4 and in a recent systematic review Howardella has been associated with depressed patients 5.

Regarding the use of relative abundances of bacterial taxa in our analyses, we indeed considered this method, however discarded this option due to the following interpretation issues of the method: 1) The interdependency of relative abundances between taxa can introduce spurious correlations, as an increase in the relative abundance of one taxon as % must result in the decrease of others, even if the absolute abundance of those taxa in reality remains unchanged; 2) The absence of  information on absolute abundances in 16sRNA sequencing: in this case the application of normalization and transformation methods in order to obtain a proxy for absolute abundancies critically relies on the chosen references. However, in clinical studies reliable ‘reference’ taxa can hardly be obtained and currently there is little empirical basis for choosing appropriate normalization factors.  These issues have been described in the following publications 6-8, among others:

On the other side, the analysis of differential abundance is statistically more complex, but it avoids the aforementioned biases and produces more informative results relative abundances when exploring differences in microbiota abundancies between groups of samples, which was the main goal of our study. The Negative-Binomial-GLM, which we have used in our DA analyses, deals well with typical distributional problems in microbiota data such as non-normal distribution and zero inflation.

Question 3: Lines 327-329: the authors make connections between Clostridia and HPA activities based on other studies. Reference 74 (Zaytsoff et al) states that corticosterone is mediated specifically by Clostridium perfringens, which is from the Clostridia class. However authors the authors erroneously connect this to their observed increase in bacterial taxa within the Clostridia class, but they do not report increases in Clostridium perfringens in the current study. Species within the same family, order, phylum etc. may have vastly different functions and I advise the authors to reassess their interpretation of microbiome data.

Answer 3. Thank you for this comment. We have since deleted this statement to avoid overinterpretation of the data and instead concentrate the association of the genus Clostridium with animal models of mental health, found in the Discussion on Page 10, Lines 334-338.

References

  1. Tang S, Chen Y, Deng F, Yan X, Zhong R, Meng Q, Liu L, Zhao Y, Zhang S, Chen L, Zhang H. Xylooligosaccharide-mediated gut microbiota enhances gut barrier and modulates gut immunity associated with alterations of biological processes in a pig model. Carbohydrate polymers. 2022;294:119776.
  2. Ribeiro PVM, Veloso TG, de Oliveira LL, Mendes NP, Alfenas RCG. Consumption of yacon flour and energy-restricted diet increased the relative abundance of intestinal bacteria in obese adults. Brazilian journal of microbiology : [publication of the Brazilian Society for Microbiology]. 2023;54(4):3085-99.
  3. Yan H, Zhao S, Huang HX, Xie P, Cai XH, Qu YD, Zhang W, Luo JQ, Zhang L, Li X. Systematic Mendelian randomization study of the effect of gut microbiome and plasma metabolome on severe COVID-19. Frontiers in immunology. 2023;14:1211612.
  4. Babayeva A, Ozkul C, Coskun M, Uzun A, Yalcin MM, Yalinay M, Akturk M, Toruner FB, Karakoc MA, Yetkin I, Altinova AE. Alteration in gut microbial characteristics of patients with acromegaly. Endocrine. 2024;85(2):855-63.
  5. Barandouzi ZA, Starkweather AR, Henderson WA, Gyamfi A, Cong XS. Altered Composition of Gut Microbiota in Depression: A Systematic Review. Frontiers in psychiatry. 2020;11:541.
  6. Bruijning M, Ayroles JF, Henry LP, Koskella B, Meyer KM, Metcalf CJE. Relative abundance data can misrepresent heritability of the microbiome. Microbiome. 2023;11(1):222.
  7. Harrison JG, John Calder W, Shuman B, Alex Buerkle C. The quest for absolute abundance: The use of internal standards for DNA-based community ecology. Molecular ecology resources. 2021;21(1):30-43.
  8. Props R, Kerckhof FM, Rubbens P, De Vrieze J, Hernandez Sanabria E, Waegeman W, Monsieurs P, Hammes F, Boon N. Absolute quantification of microbial taxon abundances. The ISME journal. 2017;11(2):584-7.

Round 2

Reviewer 2 Report

Comments and Suggestions for Authors

Thank you for the response from the authors. Regarding the relative abundance argument, this is fine, however then the authors should also display absolute abundance levels. Either relative or absolute abundances for the major taxa should be published to ensure transparency and comparability between publications.

It is not sufficient that the authors state they reanalyzed the data. How was the data reanalyzed? What was done different? This needs to be scrutinized before ready for publications. 

Author Response

Reviewer Comment 1:

Thank you for the response from the authors. Regarding the relative abundance argument, this is fine, however then the authors should also display absolute abundance levels. Either relative or absolute abundances for the major taxa should be published to ensure transparency and comparability between publications.

Author Answer 1: Thank you for this comment. For transparency, we have added Suppementary Figure 1 containing information on the relative abundance of the gut microbiota in each of our groups.

This information has been included in the Methods on Page 4 Lines 199-202, and in the Results on Page 8, Lines 270-271 and Lines 274-275.

Reviewer Comment 2:

It is not sufficient that the authors state they reanalyzed the data. How was the data reanalyzed? What was done different? This needs to be scrutinized before ready for publications.

Author Answer 2:

Our apologies for not being clear. In our previous revision, statistical reanalysis of the data was performed at the request of Reviewer 1, who asked us to include age as a covariate in our model. This resulted in slight alterations in the significance of the bacterial genera between groups. The results of including age as a covariate in the analysis of alpha and beta diversity as well as differential and relative abundance, are shown in in the manuscript in the following places: Methods: Page 4 Lines 196-199 and Results: Table 3 (Page 6-7), Figure 1 (Page 7), Table 4 (Page 7-8), Table 5 (Page 8), Table 6 (Page 8-9), Figure 2 (Page 9) and Supplementary Figure 1.